# Tick-Borne Encephalitis Virus (TBEV) Infection in Two Horses

**DOI:** 10.3390/v13091775

**Published:** 2021-09-06

**Authors:** Theresa Maria Conze, Zoltán Bagó, Sandra Revilla-Fernández, Jürgen Schlegel, Lutz S. Goehring, Kaspar Matiasek

**Affiliations:** 1Equine Medicine and Reproduction, Centre for Clinical Veterinary Medicine, Faculty of Veterinary Medicine, Ludwig-Maximilians University Munich, Veterinärstraße 13, 80539 Munich, Germany; theresa.maria@gmx.de; 2Institute for Veterinary Disease Control Mödling, Austrian Agency for Health and Food Safety GmbH (AGES), IVET, 2340 Mödling, Austria; zoltan.bago@ages.at (Z.B.); sandra.revilla-fernandez@ages.at (S.R.-F.); 3Department of Neuropathology, Institute of Pathology, School of Medicine, Technical University Munich, Trogerstraße 18, 81675 Munich, Germany; schlegel@TUM.de; 4Section of Clinical and Comparative Neuropathology, Centre for Clinical Veterinary Medicine, Ludwig-Maximilians University, Veterinärstraße 13, 80539 Munich, Germany

**Keywords:** encephalopathy, Borna disease virus (BoDV), West Nile virus (WNV), in situ hybridization, clinical case

## Abstract

A final diagnosis in a horse with clinical signs of encephalopathy can be challenging despite the use of extensive diagnostics. Clinical signs are often not pathognomonic and need to be interpreted in combination with (specific) laboratory results and epidemiological data of the geographical region of the origin of the case(s). Here we describe the diagnostic pathway of tick-borne encephalitis virus infection in two horses using established molecular diagnostic methods and a novel in situ hybridization technique to differentiate between regionally important/emerging diseases for central Europe: (i) hepatoencephalopathy, (ii) Borna disease virus, and (iii) West Nile virus infections.

## 1. Introduction

Tick-borne encephalitis virus (TBEV) or in German-speaking countries also called FSMEV (Fruehsommer-Meningoencephalitis Virus) belongs to the genus *Flavivirus* of the family *Flaviviridae* and can induce encephalitis in various mammalian hosts such as humans, deer, dogs, and sheep [1]. Similar to other flaviviruses, TBEV is a spherical, enveloped, positive-sense, single-stranded RNA virus, which was first described in 1931 in Austria [2,3]. More recently, the European Centre for Disease Prevention and Control (ECDC) reported over 12,000 human TBE cases between 2012 and 2016, which emphasizes the importance of this viral disease for central Europe [4]. The main route of infection is an infected tick bite, mainly by members of *Ixodes* spp. An alimentary route of infection has rarely been described via consumption of contaminated dairy products (goat cheese) [5]. Reservoir hosts are rodents, insectivores, and sylvatic carnivores (*Mustela*, *Vulpes*, etc.). Of the existing three virus subtypes, only the European subtype, transmitted by *Ixodes ricinus*, is seen in countries of central Europe [6]. These strains typically trigger a biphasic disease course with an initial viremia with fever, headache, malaise, and myalgia. In up to 30% of viremic humans the neuroinvasive phase starts within a few days. The main clinical signs are high fevers, severe headaches, nausea and changes in mentation as indicators of a meningo-encephalomyelitis, which can be fatal in <2% of the cases [7,8]. In Germany, human TBE cases are concentrated in the southern parts [9,10]; however, only recently TBEV-RNA was amplified from ticks in Lower Saxony in Northern Germany with close similarity to a strain endemic in Poland [11]. Austria is an endemic region where a total of 154 human TBE cases were reported only in 2018 [12]. On the other hand, TBE has been described only in few animal species worldwide. It has been confirmed in dogs, monkeys, and in horses (Austria) [13,14,15,16,17]. There is a high seroprevalence in ruminants without clinical disease, which makes them valuable indicators in surveillance programs. Despite a 23.4% seroprevalence in equids in certain regions of Southern Germany, confirmed clinical cases of TBE have not yet been conclusively described [18].

## 2. Case Presentations

### 2.1. Case 1—Germany

In June 2018, a 15-year-old Haflinger gelding was admitted to the Equine Hospital of Ludwig-Maximilians University, Munich. The gelding had a brief history of acute onset of ataxia, behavioral changes, and reduced consciousness. The referring veterinarian reported that the gelding was found in lateral recumbency a few hours before admission in pasture at a location south of Munich, Bavaria. The horse had not been under close observation for about 2 weeks as the owner was residing abroad. On clinical examination the horse was mildly tachypneic (28 breaths per minute) and with a rectal temperature of 38.0 °C (normal 37.2–38.3°). On neurological examination, the gelding displayed stupor and 4-limb ataxia (grade 4/5 on the Mayhew scale), while the horses had difficulties initiating and stopping its walking activities [19]. Cranial nerve testing was normal except for a bilaterally lack of a menace response. Serum biochemistry revealed a slight increase in creatinine kinase (486.4 U/L; reference value < 283 U/L) and LDH (715 U/L; reference value < 400 U/L) activity. Serum liver and kidney parameters, electrolyte concentrations, and hematology results were all within normal limits. Venous ammonia concentration was normal at 35 mmol/L. Cerebrospinal fluid (CSF) samples were collected via C1–C2 (lateral) centesis in the (sedated) standing horse [20]. Cell count (2 WBC/µL, reference < 5 WBC/µL), cytological evaluation and a total protein concentration (0.57 g/L, reference < 0.8 g/L) were normal. As hepatoencephalopathy was ruled out by normal blood biochemistry results, and as Borna disease virus (BoDV) infection is an important differential diagnosis under our specific epidemiological circumstances in addition to the horse’s clinical presentation, serum and CSF samples were sent to the Virology Institute of Justus-Liebig University, Giessen, Germany, for antibody detection and quantification. As clinical signs and epidemiology did not fit an Equid Herpesvirus type 1 (EHV-1)-infection with typically signs of myelopathy, diagnostics were not pursued.

Meanwhile, the gelding was treated for encephalopathy of unknown cause with flunixin meglumine (Flunidol RPS 50 mg/mL, CP-Pharma, Burgdorf, Germany) 1.1 mg/kg bwt once daily i.v., dexamethasone-Na phosphate (Rapidexon 2 mg/mL, Albrecht GMBH, Aulendorf, Germany) 0.05 mg/kg bwt once daily i.m., and with trimethoprim sulfonamides (Trimetotat 400 mg/mL, 80 mg/mL, aniMedica-international GmbH, Frankfurt, Germany) 5/25 mg/kg bwt BID p.o. While the horse was able to chew and swallow normally, access to water and food was not limited. During the next 24 h the gelding’s condition deteriorated, as he frequently lost his equilibrium. He was euthanized on day 3 of hospitalization due to convulsive activity.

A complete postmortem examination was carried out and the brain was removed for extensive investigation [21]. On necropsy and macroscopic evaluation there were no gross changes. Brain and spinal cord were fixed routinely in 10% neutral buffered formalin. After complete fixation, the brain was dissected using a standardized protocol including samples from brainstem, cerebellum, diencephalon, and telencephalon. Additional sections were collected from multiple segments of the entire spinal cord length. CNS slabs were processed automatically, embedded in paraffin and sectioned at 4 μm thickness, and mounted on slides, then stained with haematoxylin and eosin (HE).

On standard slides, subtle but widespread lymphocytic angiocentric and neuronocentric infiltrates were seen throughout grey matter of brain and spinal cord, accompanied by focal microgliosis and neuronophagia (Figure 1 panels A–B). Overall low severity changes were most extensive in cerebral cortex (frontal and parahippocampal) as well as in spinal grey matter. Viral inclusion bodies were not seen, and immunohistochemistry (IHC) of affected areas and hippocampus were negative for BoDV antigen.

Moreover, few days after euthanasia the test results for BoDV antibody concentration were received and were negative for BoDV antibody in both serum and CSF. As the histo-pathological changes were reminiscent of a flavivirus encephalomyelitis pattern, and BoDV diagnostics were negative, cut sections were sent to AGES IVET Mödling, Austria, for further flavivirus diagnostics. Since the specific West Nile virus (WNV) quantitative RT-PCR was negative, a pan-flavivirus conventional RT-PCR was performed [22,23]. The FFPE sample received was positive by the pan-flavivirus PCR and identified as TBEV by sequencing of the amplicon. Thereafter, we selected primers and probe for a TBEV (European lineage) RT-PCR. This assay confirmed the presence of nucleic acid in affected brain samples.

Based on these results, IHC for TBEV antigen and RNAScope™ in situ hybridization for TBEV RNA were carried out on PCR positive sections following protocols established for BoDV [24]. Briefly: RNAScope™ probes for in situ hybridization detecting viral RNA were ordered and performed by RNAScope™ technology as previously described using the RNAScope R (Red) 2.5 Detection Kit and the TBEV detection probe (Cat.no. 575601) (all: Advanced Cell Diagnostics, Bio-Techne, Minneapolis, MN, USA). Subsequently, deparaffinization, pretreatment and hybridization were performed according to the manufacturer’s protocol using the provided pretreatment solutions and wash buffer. All incubation steps were performed in a humidity control tray and a HybEZTM thermal unit (Advanced Cell Diagnostics, Bio-Techne, Minneapolis, MN, USA). Following hybridization, a signal was detected using Fast Red as chromogen provided by the manufacturer (RedB:RedA 1:60 ratio). Counterstaining was performed using 50% Gill’s Hematoxylin 1 (American MasterTech, Lodi, CA, USA), followed by bluing with tap water and 0.02% ammonium hydroxide water. Slides were air dried and mounted with Xylene and EcoMount (EcoMount, Biocare Medical, Concord, CA, USA). Positive hits in the case were infrequent, but distinct. Positive signals were confined to neuronal cell bodies (Figure 1C–E).

In summary, a diagnosis of TBEV-induced lymphocytic encephalomyelitis was made by defining an encephalopathy based on the clinical appearance and exclusion of other relevant diseases of the area, finally confirmed by the detection of TBEV RNA via RT-PCR and RNA-Scope in situ hybridization.

### 2.2. Case 2—Austria

In May 2019, a 9-year-old Arabian gelding was presented to a local practitioner in northeastern Austria (Vorarlberg) with acute neurological gait anomalies. The gelding reportedly had a mild fever at the same time, but all other vital parameters were within normal limits. Apparently, gait analysis revealed severe ataxia (grade 4/5) in fore and hind limbs and he was not able to initiate motion. The horse was treated with glucocorticosteroids and antimicrobial drugs (details unknown to us). However, the condition of the horse deteriorated quickly, it became recumbent and was euthanized for ethical reasons. The (head of the) animal was submitted to the Austrian national reference laboratory for equine encephalomyelitis at AGES IVET Mödling, Austria, for postmortem examination. Necropsy and CNS dissection, sampling and tissue processing were carried out as in case 1. Findings were as follows: the brain, and in particular the brain stem, showed moderate and diffuse non-purulent encephalitis with predominately lymphocytic-histiocytic (perivascular) infiltrates. Oligofocally a diffuse gliosis was seen in the cerebellum’s molecular layer. In addition, infrequent oligofocal glial nodules and neuronophagia were observed in the mesencephalon. In addition, infrequent oligofocal glial nodules and neuronophagia were observed in the mesencephalon. Similar diagnostics as in case 1 were applied. Additional sections underwent immunohistochemistry staining for BoDV antigen, which is also an endemic disease in this particular part of Austria, and TBEV testing using RT-PCR was initiated. After confirmation of the TBEV infection by RT-PCR, RNAScope™ testing was initiated and findings were identical to case 1. Again, the lymphocytic/histiocytic encephalomyelitis was associated with presence of TBEV RNA in the affected tissue, and concurrent BoDV infection was excluded via IHC and by specific BoDV-1 RT-PCR [25,26].

## 3. Discussion

Case 1 is the first report of a confirmed TBEV infection (antigen positive) causing fatal neurological disease in a horse in Germany. Two previous fatal cases in Austria have been described earlier by Luckschander et al. in 1999 [17]; however, the diagnosis at that time was based on clinical signs, serum antibody detection, and histopathology only, as IHC for TBEV was negative (Weissenböck 2021: personal communication).

Thus, the current cases are the first confirmed TBEV-cases in Germany and Austria using RT-PCR assays and novel in situ hybridization technique (RNAScope™) for a definitive diagnosis and localization of viral RNA in PCR-positive tissues.

Clinical signs of TBE in horses mainly depends on the affected brain region(s) and is therefore quite unspecific and may mimic other diseases. Considering clinical presentation and disease progression, geographical localization, season, and ante mortem laboratory results, a preliminary list of differential diagnoses was composed. This short-list included BoDV, hepatoencephalopathy and ‘other, less likely’ diseases, among which WNV, following the ‘common things are common’-principle. EHV-1 as a cause for Equid Herpesvirus-associated Myeloencephalopathy or EHM was not included, as it did not fit into any of the cases’ profiles because of the following reasons: (i) single animal affected; (ii) no recent additions to a small herd; (iii) profound clinical signs of an encephalopathy (highly unlikely for EHV-1/EHM), (iv) CSF results, (v) season (and breed) [27].

Borna disease virus (BoDV) infection was considered an important differential diagnosis as both regions, Bavaria in Southern Germany (Case 1) and Vorarlberg in Austria (Case 2), are endemic areas for BoDV [26]. In addition, both cases were presented in June and May, respectively, which fit the seasonal pattern of this disease, with a higher incidence in spring and early summer. BoDV is an enveloped, negative sense single-stranded RNA virus which can cause a progressive meningoencephalitis in horses and other mammals [28]. More recent, fatal cases of encephalitis in humans were linked to BoDV, which led to revived interest in this disease [24]. The bicolored white-toothed shrew, *Crocidura leucodon*, is likely the natural reservoir host in endemic areas [29] and transmission into the horse most likely occurs via the nasal mucosa and through the olfactory nerve and finally into the brain [30]. Similar to TBEV, immunopathologic mechanisms, with special regard to CD4+ and CD8+ T cells, are thought to play an important role in neuronal damage and finally brain atrophy [31]. Corresponding histopathologic findings are perivascular lymphohistiocytic cuffing, microgliosis, neuronal necrosis and intranuclear inclusion bodies (Joest-Deegen bodies) [32]. Diseased horses may display progressive neurological deficits over a period of 7 to 12 days, often starting with subtle changes of slow-motion eating and mastication difficulties. Alterations in mental status, i.e., somnolence and stupor, as well as cranial nerve deficits, circling and propulsive walking, ataxia and other proprioceptive abnormalities might occur [28]. Cerebrospinal fluid analysis usually reveals a lymphocytic/monocytic pleocytosis as to be expected in many but not all cases of viral encephalitis. Since the endemic area and the clinical presentation of encephalopathy were consistent with a possible (late stage) BoDV infection; further diagnostics were initiated in the first case. As reliable antigen testing is not available, testing of serum and CSF antibody concentrations in addition of testing blood-brain barrier (BBB) integrity, represents an appropriate alternative option as intravitam diagnosis (sensitivity 88%, specificity 100%). Considering a 10% seroprevalence of BoDV antibodies in normal horses in endemic areas, CSF analysis for antibody detection is crucial for a valid diagnosis. Although the immunofluorescence assay did not detect BoDV antibodies in CSF or serum, a definitive diagnosis can only be made via postmortem IHC, or PCR testing of brain tissue [33,34]. Results were not indicative for BoDV detection in either animal, although PCR-testing in case 1 was not pursued.

In horses with severe liver disease and thereby impaired liver function or hepatic shunts, ammonia accumulates in the blood stream and can induce a hepatoencephalic syndrome. Increased blood ammonia may also develop in cases of bacterial enterocolititis with an overgrowth of ammonia producing microorganisms [35]. Ammonia can diffuse across the BBB and has an indirect effect on neurons due to increased oxidative stress as well as a direct toxic effect [36,37]. In addition, hyperammonemia increases the amount of the inhibitory neurotransmitter gamma-aminobutyric acid (GABA), and it also induces the accumulation of the major excitatory transmitter glutamine within astrocytes. This results in astrocyte swelling and cerebral edema [38]. A histopathological clue for hepatoencephalopathy is Alzheimer-Type 2 changes in the astrocytes. Affected horses may show mental alterations and mood swings between different levels of consciousness, starting with somnolence in the early stages followed by aggressive behavior later and finally coma or convusive activity. Ataxia and cortical blindness have also been described in these horses [39,40]. The normal blood values for ammonia and liver enzyme activities as well as the missing signs of enterocolitis made hepatoencephalopathy highly unlikely as a differential diagnosis for either case.

Despite a high seroprevalence of 23.4% for TBEV antibodies in Southern Germany, proven clinical cases of neurological disease caused by TBEV infection are lacking [18]. Due to the high seroprevalence in healthy horses, the definitive, reliable diagnosis of TBEV infection in horses with neurological clinical signs can currently only be made if other probable causes are excluded and supported through seroconversion testing and, finally, via postmortem antigen detection. The main permanently infected vector for TBEV is the tick *Ixodes ricinus* with three blood-feeding stages: larva, nymph, and adult stages. Within the tick population the virus is transmitted transstadial and transovarial, as well as during co-feeding of ticks on the same host [41]. Virus persistence in endemic areas depends on horizontal transmission from an infected tick to a susceptible vertebrate host and vice versa [42].

Natural reservoir hosts for TBEV are mainly rodents and insectivores, as well as small (wildlife) carnivores. Due to the long-lasting viremic periods, these hosts form an important virus reservoir [43]. In contrast, large mammals such as deer, goats, sheep, and cattle are considered indicator hosts for TBE virus occurrence with only brief viremic periods [42,44]. TBE virus usually enters the host organism via tick bite and the first replication takes place in residential Langerhans cells or APCs. The virus is transported to the regional lymph nodes, where another period of viral replication occurs. Thereafter, the virus enters the blood stream (first viremia) and disseminates into multiple organs (e.g., liver, spleen). From there, a CNS invasion can occur during a second round of viremia. With obvious clinical signs of neurological disease in both cases, the virus apparently became neuroinvasive. The exact mechanism of BBB entry is not fully understood, but high levels of viremia are necessary for successful BBB crossing [45,46]. The permeability of the BBB increases when high virus concentrations already exist in the brain and neurological signs are even evident. Therefore, the destruction of the BBB seems to be a secondary consequence of brain infection rather than the primary route of viral entrance. Once the virus enters the brain, the primary targets for active virus replication are neurons and to a lesser extent, astrocytes. Neuronal damage may be directly induced by viral invasion, but destruction is also triggered by inflammatory cytokines, activated microglia and immune cells such as CD8+ T-cells [47,48,49,50]. These facts emphasize that the immunopathology takes an important role in development and severity of neuronal injury caused by TBEV. The histopathology of brain slices of both cases corresponds to the earlier described neuronal alterations with lymphocytic infiltrations, satellitosis, neurodegeneration, and neuronophagia by glial cells. In both cases the grey matter was most affected, as is typical for flavivirus infections [51]. The inflammatory accentuation in the spinal cord, brainstem, and cerebellum has been described, however, signs of inflammation in the latter two areas were low. Interestingly, inverse topographical correlation of inflammation and detectability of viral antigen was observed in a study by Gelpi et al. (2005) [16,52]. In their retrospective study, the distribution of TBEV in human brains was analyzed using IHC, and the distribution pattern was compared with results of histopathology and serology. These authors determined a maximal detectability of viral antigen via IHC between days 6 and 13 of disease. If durations of disease and clinical symptoms exceeded 4 weeks, IHC became inadequate as a diagnostic, while TBEV PCR results remained positive. Applying this to our two equine cases, the longer duration of disease and a rapid virus clearance could be a possible explanation for the negative IHC results in multiple sections, despite positive TBEV PCR results. In addition, in situ hybridization positive signals were sparse and restricted to few neuronal cell bodies, which further shows a possible retreat and clearance of viral antigen with time. Intra-vitam diagnosis of TBE seems to be challenging, since viral antigen can only be detected in serum and CSF during the first viremic phase of infection [1,53]. The virus has already been cleared from the blood and CSF when neurological symptoms occur, and specific antibodies are generated. Therefore, the diagnosis in humans is mainly based on the detection of early-produced antigen specific IgM- and IgG-antibodies or seroconversion. Previous vaccination in humans and cross-reactions with other flaviviruses may interfere with the serologic test results, which therefore should be confirmed with virus neutralization (VN) assays. This procedure is applied in the horse, where an established WNV ELISA in a positive case should be followed by specific WNV VN-testing. Once the VN-testing is negative, a TBEV infection as a related Flavivirus member is likely.

WNV, which is a commonly found flavivirus infection in horses in specific areas of the world, was included in the differential diagnosis; however, it was considered less likely due to epidemiological reason. To date, a WNV diagnosis in an equine case has not been reported in either region of case origin, and moreover, ‘season’ or ‘time of year’ did not fit with a WNV infection, as peak season for both, humans and horses is late summer to early fall. Mosquitoes are the vector for this virus [54]. WNV was first isolated in Africa in 1937 [55]. It reached the Mediterranean basin in 1960, emerged at the East Coast of North America in 1999, and became endemic in most states (USA) or provinces (Canada) of North, Central, and South America by 2005. WNV emerged in Eastern Austria (Vienna area) in 2016, presumably via a Balkan route, and then did emerge in Germany (Saxony) in the fall of 2018 [56]. Certain bird species are natural reservoirs for WNV with high viral loads and long viremic periods, whereas humans and horses are considered dead-end hosts with a short-duration (hours–days) viremia [57]. About 10% of the infected equids develop unspecific clinical signs such as fever and anorexia, which can be followed by variable neurologic deficits [58,59]. More commonly, affected horses display hyperaesthesia and irritability; muscle fasciculations especially of the head; gait abnormalities including ataxia and weakness, as well as behavioral alterations and cranial nerve deficits [60]. The clinical course apparently is quite similar to TBE infection. Serological differentiation between TBEV- and WNV- infection is challenging, since both are members of the *Flavivirus* genus, and WNV, dengue virus, yellow fever virus, Japanese encephalitis virus (JEV) and TBEV are structurally highly similar and serological cross-reactivity is common [61,62]. Screening tests using commercial ELISA for diagnosis of WNV infections in equine blood serum could therefore lead to false positive results [63]. The suspicion of WNV infection needs to be confirmed by time- and cost-consuming virus-specific neutralization assays. Recently, Rockstroh et al. (2019) developed a new diagnostic ELISA to differentiate between TBEV and WNV in human and equine blood serum. The group generated mutant envelope proteins, which lack conserved parts of the predominant target for cross-reacting antibodies. These proteins were incubated with equine and human sera with known infections of various flaviviruses. Specificity and sensitivity in horses were increased to 100% and 87.9%, respectively, for WNV, and 95% and 96.7% for TBEV. This promising approach might improve specificities in serologic differentiation of flaviviruses in the future [64].

## 4. Conclusions

Neuropathology caused by TBEV occur in horses living in endemic regions. However, the more common differential diagnoses needed to be excluded. Because of high seroprevalence and cross-reactivity with viral antibodies raised against other flaviviruses, the intra vitam diagnosis of TBE is challenging in horses, but diagnostic sensitivity will advance as WNV cases have been conclusively detected in (Eastern) Germany, and the broader knowledge on WNV diagnostics will become available.

## Figures and Tables

**Figure 1 viruses-13-01775-f001:**
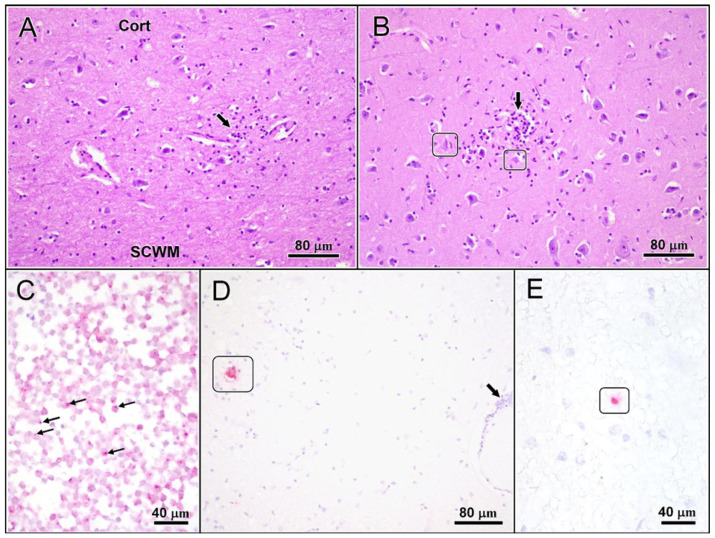
(panels **A**–**E**): Brain changes of TBEV-infected 15-year-old haflinger gelding. (**A**,**B**): Histological changes comprise multifocal lymphohistiocytic infiltration and microgliosis (arrows) scattered throughout grey matter of cortices (**A**) and nuclei (**B**). The infiltrates are occasionally associated with nerve cell degeneration (frames) and neuronophagia (not shown) indicating cytopathic activity of the virus. (**C**–**E**): Viral RNA has been visualized using RNA Scope^®^ technology in infected HeLa cells (**A**: small arrows), used as the positive control, and in affected brains (**D**,**E**). At this stage of infection, the yield of viral RNA flags up in individual pyramidal (**D**: frame) and granule neurons (**E**: frame) only, even distant from other tissue changes such as lymphocytic perivascular infiltrates (Barrow). Cort: cortex; SCWM: subcortical white matter. Stains/chromagens: (**A**,**B**): haematoxilin-eosin; (**C**–**E**): haematoxilin (blue)/AEC (red).

## Data Availability

Patient administration and case logbooks of referring veterinarians and participating institutions.

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
