# Peer review of "Tick-Borne Encephalitis Virus (TBEV) Infection in Two Horses"

_viruses, 2021, doi:10.3390/v13091775_

Round 1
Reviewer 1 Report
This is a nice description of 2 cases of TBEV in horses in Germany and Austria.
I have only minor edits/comments:
P2. Line 69 - Delete the “(Pease et al.2012)”
P2. Lines 79, 80 – add brand name and company
P3. Line 142 – put a dot (.) and space between “case1 Findings”
Figure 1. It is not clear from which TBEV case those slides were generated. There are no references in the text to Figure 1.
P5. Line 218. Replace Horses with “In horses”
P6. Lines 258-260. The sentence starting with “At this point … “. It is not clear to me why “the destruction of BBB seems to be a secondary consequence of brain infection rather than the primary route of viral entrance.” What is the primary route of viral entry into the brain then? Please clarify.
Author Response
Dear Reviewer,
thank you for your kind word and for you valuable comments. We made all the changes and rephrased the section on BBB pathology and entry. The latter is still unclear, and we rephrased our previous wording.
P2. Line 69 - Delete the “(Pease et al.2012)” -done
P2. Lines 79, 80 – add brand name and company -done
P3. Line 142 – put a dot (.) and space between “case1 Findings” -done
Figure 1. It is not clear from which TBEV case those slides were generated. There are no references in the text to Figure 1. - all images are from case 1. However, as we describe, findings in the 2 horses were very similar.
P5. Line 218. Replace Horses with “In horses” -done
P6. Lines 258-260. The sentence starting with “At this point … “. It is not clear to me why “the destruction of BBB seems to be a secondary consequence of brain infection rather than the primary route of viral entrance.” What is the primary route of viral entry into the brain then? Please clarify. - we rephrased. Our main arguments are that CSF analysis of the horse with clinical signs did not reflect BBB breeching or damaging.
Thank you for your valuable comments and suggestions.
Reviewer 2 Report
In the presented manuscript, the authors described two horses with TBEV infection. The topic is interesting, since TBEV is one of the most widely distributing flaviviruses and clinical cases in horses are rarely detected. Please see some minor comments below.
Abstract
Please explain abbreviations.
Keywords: please delete Fruehsommer Meningoenzephalitis virus (FSMEV).
Introduction
Line 32: please correct "epidemiological significance of this viral disease"
Line 37: please correct Ixodes ricinus.
Line 48: please correct "seropositive for TBEV without clinical ..."
Case presentations
Line 70: please add reference values for the CSF protein level.
Lines 79 and 80: brand name and company are missing in the parentheses.
Discussion
Discussion is too long. I suggest authors to shorten this section (lines 189-196, 245-255, 299-307) and discuss the results of the study in comparison with similar studies.
Line 179: please correct hepatoencephalopathy.
Line 184: to my knowledge EHV-1 shows no seasonal distribution and should be included in the differential diagnosis.
Line 236: please rephrase "clinically normal horses".
Line 237: please correct " ... TBEV infection in horses with neurological symptoms can currently only ... "
Line 240: please correct Ixodes ricinus.
Line 280: please correct TBEV RT-PCR results.
Line 293: please delete West Nile virus, abbreviation is explained above.
Line 299: please correct Japanese encephalitis.
Lines 313-314: please correct dengue virus, yellow fever virus, Japanese encephalitis virus.
Author Response
Dear Reviewer, thank you for your time and for your valuable comments and suggestions. We corrected accordingly.
Abstract
Please explain abbreviations. -done
Keywords: please delete Fruehsommer Meningoenzephalitis virus (FSMEV). -done
Introduction
Line 32: please correct "epidemiological significance of this viral disease" -we rephrased
Line 37: please correct Ixodes ricinus. -done
Line 48: please correct "seropositive for TBEV without clinical ..."
-done
Case presentations
Line 70: please add reference values for the CSF protein level. -done (<800mg/L)
Lines 79 and 80: brand name and company are missing in the parentheses. -done
Discussion
Discussion is too long. I suggest authors to shorten this section (lines 189-196, 245-255, 299-307) and discuss the results of the study in comparison with similar studies. -deleted
Line 179: please correct hepatoencephalopathy. -done
Line 184: to my knowledge EHV-1 shows no season al distribution and should be included in the differential diagnosis.
With all due respect, having a strong background in EHV-1 epidemiology and pathobiology, we changed the reference. While incidental EHM outbreaks are seen during the summer months, the majority of outbreaks occurs during the months of November to May (northern hemisphere). Furthermore, for us as clinicians the profound clinical signs of an encephalopathy in the 2 cases, did not fit with the profile of EHV-1 as a cause, as those signs are of a myelopathy with undisturbed mentation or brain stem signs.
Line 236: please rephrase "clinically normal horses". -done
Line 237: please correct " ... TBEV infection in horses with neurological symptoms can currently only ... "
Line 240: please correct Ixodes ricinus. -done
Line 280: please correct TBEV RT-PCR results. -done
Line 293: please delete West Nile virus, abbreviation is explained above. -done
Line 299: please correct Japanese encephalitis. -done
Lines 313-314: please correct dengue virus, yellow fever virus, Japanese encephalitis virus. -done
Thank you again for your valuable time!